# Hematopoietic Stem Cells as an Integrative Hub Linking Lifestyle to Cardiovascular Health

**DOI:** 10.3390/cells13080712

**Published:** 2024-04-19

**Authors:** Xinliang Chen, Chaonan Liu, Junping Wang, Changhong Du

**Affiliations:** State Key Laboratory of Trauma and Chemical Poisoning, Institute of Combined Injury, Chongqing Engineering Research Center for Nanomedicine, College of Preventive Medicine, Third Military Medical University, Chongqing 400038, China; 18323060623@163.com (X.C.); naomiliu0306@126.com (C.L.)

**Keywords:** hematopoietic stem cell, myeloid bias, cardiovascular disease, lifestyle

## Abstract

Despite breakthroughs in modern medical care, the incidence of cardiovascular disease (CVD) is even more prevalent globally. Increasing epidemiologic evidence indicates that emerging cardiovascular risk factors arising from the modern lifestyle, including psychosocial stress, sleep problems, unhealthy diet patterns, physical inactivity/sedentary behavior, alcohol consumption, and tobacco smoking, contribute significantly to this worldwide epidemic, while its underpinning mechanisms are enigmatic. Hematological and immune systems were recently demonstrated to play integrative roles in linking lifestyle to cardiovascular health. In particular, alterations in hematopoietic stem cell (HSC) homeostasis, which is usually characterized by proliferation, expansion, mobilization, megakaryocyte/myeloid-biased differentiation, and/or the pro-inflammatory priming of HSCs, have been shown to be involved in the persistent overproduction of pro-inflammatory myeloid leukocytes and platelets, the cellular protagonists of cardiovascular inflammation and thrombosis, respectively. Furthermore, certain lifestyle factors, such as a healthy diet pattern and physical exercise, have been documented to exert cardiovascular protective effects through promoting quiescence, bone marrow retention, balanced differentiation, and/or the anti-inflammatory priming of HSCs. Here, we review the current understanding of and progression in research on the mechanistic interrelationships among lifestyle, HSC homeostasis, and cardiovascular health. Given that adhering to a healthy lifestyle has become a mainstream primary preventative approach to lowering the cardiovascular burden, unmasking the causal links between lifestyle and cardiovascular health from the perspective of hematopoiesis would open new opportunities to prevent and treat CVD in the present age.

## 1. Introduction

Cardiovascular disease (CVD) comprises a group of diseases, including coronary heart disease, cerebrovascular disease, rheumatic heart disease, and peripheral arterial vascular disease, that develop in the heart and blood vessels. Atherosclerosis, which is an inflammatory disease of the large arteries, is the main pathological basis of CVD [1]. CVD has been the leading cause of disability and death in the global population, accounting for approximately one-third of all deaths [2]. Despite increasing awareness of CVD and breakthroughs in modern medical care, CVD incidence continues to rise year by year and shows a trend of affecting younger individuals [3]. In epidemiological studies, modifiable factors including systolic blood pressure, body mass index, tobacco smoking, low-density lipoprotein cholesterol level, and diabetes, most of which are closely associated with lifestyle, have been consistently linked to enhanced cardiovascular risk [4], underscoring the pivotal implications of lifestyle in CVD pathogenesis and management.

The current lifestyle has changed drastically from that of decades ago due to complex changes in the socioeconomic environment. Alarmingly, the modern lifestyle gives birth to many emerging cardiovascular risk factors including psychosocial stress, sleep problems, physical inactivity/sedentary behavior, alcohol consumption, tobacco smoking, and metabolic disorders [5]. An increasing number of studies propose that the modern lifestyle underpins the paradox of the prevalence of CVD and scientific/technological progress. Correspondingly, adhering to a healthy lifestyle has become a mainstream primary preventative approach to lowering the cardiovascular burden [6]. Nevertheless, the causal links between lifestyle and cardiovascular health remain mysterious; unmasking them would open new opportunities to prevent and treat CVD in the present age.

The hematological and immune systems are positioned at the interface of the interplay between external stimuli and internal pathophysiological responses. In particular, myeloid leukocytes have been well recognized as a pivotal pathological basis for CVD [7]. Myeloid leukocytes are short-lived (~3 days) in circulation and must be continuously replenished by hematopoietic stem cells (HSCs), which primarily reside in the bone marrow (BM). Interestingly, recent studies show that alterations in BM HSC homeostasis, which result in the persistent overproduction of pro-inflammatory myeloid leukocytes, occur ahead of and underpin CVD pathogenesis [8]. Meanwhile, accumulating studies including our own have demonstrated that BM HSCs are particularly vulnerable to lifestyle changes [9,10,11]. These lines of evidence suggest that BM HSCs may act as an integrative hub through which lifestyle signals are sensed and relayed to the cardiovascular system. Here, we review recent advances in elucidating the mechanistic interrelationships among lifestyle, HSC homeostasis, and cardiovascular health.

## 2. Overview of HSC Homeostasis

HSCs were originally defined as a population of cells with the ability to continuously self-renew and maintain their full-lineage differentiation potential throughout their lives. HSCs are rare and quiescent in BM, and their homeostasis is stably maintained at steady state by the delicate coordination of self-renewal, differentiation, mobilization/homing, and death/survival [12]. Traditionally, it is believed that HSCs produce blood cells in a stepwise manner, strictly adhering to a tree-like hematopoietic hierarchy in which HSCs are phenotypically and functionally considered a relatively homogeneous population [13]. In this classical hematopoietic hierarchy, HSCs initially undergo differentiation into multipotent progenitors (MPPs) with a restricted self-renewal capacity. Then, the first lineage separation occurs between common myeloid progenitors (CMPs) and lymphoid progenitors (CLPs) downstream of the MPPs. CMPs generate granulocyte–monocyte progenitors (GMPs) and megakaryocyte–erythroid progenitors (MEPs), while CLPs produce lymphocytes and dendritic cells. GMPs further differentiate into macrophages, granulocytes, and dendritic cells, whereas MEPs give rise to erythrocytes and megakaryocytes, the precursors of platelets (Figure 1A).

However, recent advances in single-cell lineage tracing and sequencing technologies have delineated a tremendous heterogeneity within HSCs at both the molecular and functional levels. Based on the molecular signature and lineage output of myeloid versus lymphoid cells, HSCs can be classified into myeloid-biased, balanced, and lymphoid-biased HSCs, with megakaryocyte (MK)/platelet-biased HSCs being identified as a subset of myeloid-biased HSCs positioned at the apex of the hematopoietic hierarchy [13,14]. In the heterogeneous HSC population, lineage-biased HSCs are distinguished by a skewed production of megakaryocytes/platelets and myeloid or lymphoid cells while retaining their multi-lineage potential, whereas balanced HSCs produce all lineages of blood cells roughly equivalently (Figure 1B). These exciting studies indicate that lineage separation actually occurs earlier than anticipated in the classical hematopoietic hierarchy, seemingly at the HSC level [14]. Moreover, tracing HSC emergence via single-cell sequencing has revealed that the lineage separation of HSCs may occur as early as the embryonic phase [15]. Notably, it was recently reported that the lineage differentiation of megakaryocytes is quite distinct from that of other lineages as HSCs’ commitment into megakaryocytes can bypass traditional lineage checkpoints and directly give rise to megakaryocytes independent of other lineages [16] (Figure 1B).

HSC homeostasis is maintained by both intrinsic and extrinsic mechanisms. Intrinsic mechanisms involve molecular signals at the epigenetic, transcriptional, and translational levels [12]. Additionally, metabolic regulation is also recognized as a powerful intrinsic principle dictating HSC homeostasis [17]. Extrinsic mechanisms are implemented by the specialized BM microenvironment, also termed a niche, in which HSCs reside. The HSC niche comprises a variety of cell types ranging from HSC descendants including megakaryocytes and immune cells to non-hematopoietic cells including osteoblasts, perivascular cells, adipocytes, nerves, and endothelial cells. These cellular components of the HSC niche regulate HSC homeostasis through physical interaction or intercellular signals in the form of secreted or cell-bound factors [18]. Emerging studies also spotlight the roles of the extracellular matrix shaped by niche cells in the regulation of HSC homeostasis [19,20]. Moreover, the nutrients sourced from circulation, which can be modified by diet, also act as niche factors and regulate HSC homeostasis through modulating nutrient sensing pathways, rewiring metabolic flux, or remodeling the cellular niche, as reviewed below. In addition, inter-organ factors, such as those sourced from the central nervous system [21], liver [22], kidney [11], and ovary [23], were recently demonstrated to be implicated in the maintenance of BM HSC homeostasis as well. Overall, the sophisticated interplays between these intrinsic and extrinsic factors synergistically keep HSC homeostasis in check.

## 3. Hematopoietic Underpinnings of CVD

Although the pathogenesis of CVD is rather complex, inflammation provoked by inflammatory cells, particularly those of the myeloid lineage, is well recognized as the most critical initiating factor for the entire spectrum of CVD [7,24]. The dichotomy of myeloid leukocytes within cardiovascular tissues, that is, recruitment from blood versus local proliferation, has been well documented. Recent conceptual and technological breakthroughs have further expanded our knowledge on the origins and actions of myeloid leukocytes in CVD. Particularly promising discoveries are the association of CVD with clonal hematopoiesis, that is, the massive production of myeloid leukocytes from a sub-clone of the HSC population such as MK/myeloid-biased HSCs in BM, and the epigenetic imprint conveyed from HSCs of a pro-inflammatory phenotype in myeloid leukocytes [7]. The pathogenic roles of clonal hematopoiesis and myeloid leukocytes in CVD have been comprehensively reviewed elsewhere [7,24,25]. Recently, accumulating studies have spotlighted the links between traditional cardiovascular risk factors and HSC homeostasis, showing that cardiovascular risk factors act on the lineage separation of HSCs, leading to MK/myeloid-biased hematopoiesis that is manifested by the overproduction of pro-inflammatory myeloid leukocytes and hyperreactive platelets, which predisposes people to cardiovascular inflammation and thrombosis [8].

Moreover, in addition to acting as a driver, MK/myeloid-biased hematopoiesis also exists as a consequence of CVD. Rohde et al. report that CVD-associated inflammation provokes the overproduction of pro-inflammatory myeloid leukocytes through remodeling the vascular niche of HSCs [26]. Meanwhile, HSCs themselves are able to sense inflammation and adapt through proliferation and MK/myeloid-biased differentiation [27]. For example, interleukin 1 (IL-1), IL-6 and tumor necrosis factor α (TNF-α) are among the most common inflammatory cytokines in CVD [24]. IL-1 activates a PU.1-dependent gene program to directly accelerate the proliferation and myeloid differentiation of HSCs [28,29,30]. IL-6 directly acts on HSCs and epigenetically reprograms HSCs through the JAK/STAT3 signaling pathway, which features imprints of myeloid and pro-inflammatory gene signatures that can be conveyed to progenies of the myeloid lineage [31,32]. TNF-α activates a strong and specific NF-κB-dependent gene program to directly promote the survival and myeloid differentiation of HSCs [33,34]. Moreover, the inflammatory IL-33–suppressor of tumorigenicity (ST2) axis, which is often dysregulated in CVD [35,36], can promote the mobilization and myeloid differentiation of HSCs as well [37]. In addition, CVD is also distinguished by its oxidative status [38]. The resultant oxidative stress may promote the proliferation and MK/myeloid-biased differentiation of HSCs through the activation of the p38 MAPK or cytosolic nucleic acid-sensing pathways [39,40], analogous to the effects in cardiovascular cells [41,42,43] and platelets [44,45,46] we reported before. In addition, increased HSC proliferation in CVD further expedites the somatic evolution and expansion of clones with driver mutations, increasing the risk of clonal hematopoiesis [47]. Overall, these tight interplays between HSC homeostasis alterations and CVD may form a vicious cycle driving the rapid progression of CVD.

## 4. CVD-Promoting Lifestyle Remodels BM HSC Homeostasis

At present, emerging cardiovascular risk factors that mainly arise from unhealthy lifestyle factors, such as psychosocial stress, sleep problems, an unhealthy diet pattern, physical inactivity/sedentary behavior, alcohol consumption, and tobacco smoking, are increasingly attracting attention [3]. Below, we review how alterations in HSC homeostasis could link these lifestyle factors to CVD pathogenesis (Table 1).

### 4.1. Psychosocial Stress

Psychosocial stress is an unavoidable element of life, afflicting all humans acutely or chronically across a range of intensities and frequencies. Psychosocial stressors can exist in many forms, including adverse socioeconomic or working conditions and life changes, all of which seriously undermine the mental health of people and are even associated with chronic psychiatric conditions such as anxiety and depression [85]. Alarmingly, epidemiological studies have shown that more than 30% of adults suffer from anxiety disorders, and this prevalence continues to grow [86]. Psychosocial stress promotes CVD pathogenesis, potentially through direct mechanisms involving the hypothalamic–pituitary–adrenal (HPA) axis-mediated production of glucocorticoids and the sympathetic–adrenal–medullary axis-mediated release of catecholamines or indirectly through behavioral mechanisms involving smoking, physical inactivity, or excessive alcohol consumption, leading to an increase in the prevalence and severity of traditional cardiovascular risk factors such as obesity, diabetes, and hypertension [87].

Like most tissues, the sympathetic nervous system (SNS) highly innervates the BM, influencing certain aspects of hematopoiesis [88]. In mouse models of psychosocial stress, chronic SNS activation not only increases the proliferation of HSCs but also mobilizes HSCs from the BM to the spleen, where they undergo extramedullary hematopoiesis; both drive the production of pro-inflammatory myeloid leukocytes, leading to the accelerated development of atherosclerosis [48,49,50]. Meanwhile, SNS activation also expands MKs in the BM, increasing the number of platelets numbers in circulation [89]. These extensive hematopoietic alterations suggest that psychosocial stress may affect lineage separation at the HSC level, promoting the MK/myeloid-biased differentiation of HSCs. However, in traumatic brain injuries, the hypersensitivity of the SNS promotes BM HSC proliferation but skews their differentiation into anti-inflammatory myeloid leukocytes [90]. The discrepancy of differentiation into pro-inflammatory or anti-inflammatory myeloid leukocytes following SNS activation suggests a dependence on the type, intensity, and duration of stress.

As far as we know, SNS activation mainly modulates HSC homeostasis through indirect mechanisms and highly relies on catecholamines and related adrenergic receptor (AR) signaling. Sympathetic receptors, especially β3-AR, are present in HSC niche cells such as mesenchymal stem cells (MSCs), which are the main source of hematopoietic retention factor CXC chemokine receptor type 12 (CXCL12/SDF1) in BM. SNS activation releases noradrenaline, which signals BM niche cells via β3-AR to downregulate CXCL12 expression [49]. Meanwhile, SNS activation also prompts neutrophils to secrete proteases to cleave CXC chemokine receptor type 4 (CXCR4), the receptor for CXCL12, on the HSC surface [50]. This diminished CXCL12–CXCR4 signaling then provokes HSC proliferation and mobilization. There is currently no evidence pointing to the direct action of catecholamines on HSCs. In fact, β3-, β2-, and α2-ARs function in hematopoietic cells. For example, we have demonstrated that α2-ARs on megakaryocytes mediate ERK activation to promote megakaryocyte adherence and migration, leading to enhanced platelet production [91]. β2-ARs are expressed on the progenitors and mature cells of the myeloid lineage; their ablation diminishes myeloid progenitor proliferation and myeloid leukocyte development [92]. Meanwhile, Liu et al. report that SNS-derived dopamine regulates HSC proliferation directly through D2-type dopamine receptors. D2-type dopamine receptor activation removes the inhibition by PKA of Lck expression, which then augments stem cell factor (SCF) signaling in HSCs [51]. Gao et al. also report that nociceptive neurons in the BM collaborate with sympathetic nerves to drive HSC mobilization via the secretion of CGRP, which acts on HSCs directly via RAMP1 and CALCRL to activate the Gαs/adenylyl cyclase/cAMP pathway [93]. Based on these lines of evidence, it is plausible that catecholamines may act directly on HSCs as well. Notably, Dong et al. recently reported that norepinephrine release during SNS activation upregulates the expression of lysine methyltransferase 5A (KMT5A), which specifically promotes the histone mono-methylation modification of spleen tyrosine kinase (SYK) in monocytes [94]. Whether catecholamines also remodel the HSC epigenome, as we discuss below, remains unknown.

Analogous to SNS activation, glucocorticoids, as stress hormones, also modulate HSC homeostasis. Pierce et al. show that hypothalamic Chrm1 signaling stimulates HSC mobilization in mice through the hormonal priming of the HPA axis to release glucocorticoids. Then, physiological levels of glucocorticoids facilitate HSC mobilization via the glucocorticoid receptor NR3C1-mediated upregulation of actin-organizing proteins [21]. Additionally, repeated social-defeat-stress-induced glucocorticoid signaling activation promotes the egress of pro-inflammatory monocytes from BM into circulation through the downregulation of BM CXCL12 expression [52]. Thus, glucocorticoid signaling-induced BM CXCL12 downregulation may also promote HSC mobilization. However, in contrast to the effect of glucocorticoids on mice, Guo et al. identified the glucocorticoid receptor as a transactivator of CXCR4 in human HSCs, the activation of which by a synthetic glucocorticoid, dexamethasone, enhances CXCL12–CXCR4 axis-mediated BM homing and the long-term engraftment of HSCs [95]. This discrepancy may reflect a dose- or context-dependent effect of glucocorticoids. Currently, there is no evidence regarding whether glucocorticoids affect the lineage commitment of HSCs.

### 4.2. Sleep Problems

Sleep is also an integral element of life. Yet sleep problems, including insufficient or disturbed sleep, are increasingly common, largely due to contemporary changes to our work or social schedules and living environments, and have becoming a growing and underappreciated determinant of health in modern society [96]. Sleep problems are now well recognized to increase the risk of multiple pathological conditions including CVD [97]. Moreover, patients with heart disease also frequently show disruptions in their sleep–wake cycles which, in turn, considerably contribute to the overall disease burden [98].

The sleep–wake cycle is a manifestation of the circadian rhythm, which is established by networks of molecular oscillators in the brain as well as in peripheral tissues interacting with environmental and behavioral cycles, including hormones (e.g., catecholamines and melatonin), the tonus in the parasympathetic and sympathetic nervous systems’ innervation of tissues, and the effect of light [99]. Thus, sleep problems disrupt the circadian rhythm, causing neuroendocrine disorders. In support of this, McAlpine et al. demonstrate that hypocretin, a wake-promoting neuropeptide secreted by the lateral hypothalamus, maintains HSC quiescence through inhibiting colony-stimulating factor 1 (CSF1) production by BM pre-neutrophils expressing the hypocretin receptor. Sleep fragmentation causes less hypocretin production, and the resultant CSF1 elevation then enhances the proliferation and myeloid-biased differentiation of HSCs, leading to the overproduction of pro-inflammatory monocytes and the development of larger atherosclerotic lesions [53]. To make matters worse, sleep deprivation also elevates levels of brain prostaglandin D2 (PGD2), the efflux of which across the blood–brain-barrier induces systemic inflammation characterized by an accumulation of pro-inflammatory cytokines and neutrophils via the G-protein-coupled receptor DP1 [54]. This inflammation may then increase the proliferation and myeloid-biased differentiation of HSCs, as described above. Moreover, catching up on sleep through napping and weekend compensatory sleep insufficiently offsets the impacts of sleep debt on HSCs as sleep interruption restructures the HSC epigenome, increasing proliferation, skewing myeloid commitment, and enhancing pro-inflammatory priming [55]. Meanwhile, the authors propose that sleep limits neutral drift to preserve HSC clonal diversity, while sleep interruption reduces HSC clonal diversity and promotes clonal hematopoiesis through accelerated genetic drift [55].

On the other hand, HSC homeostasis also adheres to a circadian rhythm featuring two daily peaks that are initiated by the onset of light and darkness. One decade ago, research revealed robust circadian fluctuations in circulating HSCs, peaking 5 h after the initiation of light and reaching a minimum 5 h after darkness. The circadian HSC release is driven by the SNS secretion of noradrenaline, which is locally transmitted to stromal cells via β3-ARs, resulting in rapid CXCL12 downregulation through reducing the nuclear content of transcription factor Sp1 [56]. The light-induced secretion of norepinephrine and TNF-α also augment HSC differentiation and increase vascular permeability in the BM to replenish peripheral blood, while darkness-induced TNF-α bursts increase the secretion of melatonin, which drives self-renewal and BM retention of HSCs through elevating BM COX-2/αSMA^+^ macrophages [57]. Additionally, García-García et al. report a dual regulation of the daily migration of HSCs and leukocytes by the parasympathetic and sympathetic nervous systems. Firstly, light-triggered sympathetic activity promotes the egress of HSCs and leukocytes from BM into circulation via two neurotransmitters, with acetylcholine suppressing vascular adhesion while norepinephrine attenuates MSC-mediated and CXCL12-dependent BM retention through β3-ARs. At night, in contrast, sympathetic cholinergic signals reduce β3-AR expression in BM, while endocrine-derived epinephrine is elevated to promote vascular adhesion and BM retention of HSCs and leukocytes in a β2-AR-dependent manner [58]. Furthermore, the diurnal egress of HSCs from BM into circulation is also reported to be regulated by the complement, coagulation, and fibrinolytic cascades, the activation of which is triggered in a circadian manner [59]. The activation of the complement cascade cleaves C5 via C5 convertase in BM sinusoids into C5a, which attenuates CXCL12–CXCR4 and VLA4–VCAM-1 retention signals through stimulating the release of several proteolytic enzymes by myeloid leukocytes in the BM [60]. Concurrently, the coagulation and fibrinolytic cascades can provide C5 convertase activity through generating thrombin and plasmin, respectively [61]. Researchers also observed a circadian activation of inflammasome signaling in HSCs which increases the autocrine release of extracellular ATP to promote CXCR4 incorporation into the membrane lipid rafts of HSCs [62,63]. Based on the above evidence, sleep problems seem to be able to disrupt the circadian behavior of HSCs, resulting in greater HSC differentiation than self-renewal as well as greater HSC mobilization than BM retention. Meanwhile, as we discussed above, the resultant exacerbation of SNS activity and neuroendocrine disorders may also activate HSCs and skew their differentiation into pro-inflammatory myeloid leukocytes. Altogether, sleep problems can cause more accumulation of pro-inflammatory myeloid leukocytes in circulation, thereby increasing the risk of CVD.

### 4.3. Unhealthy Diet Pattern

Changes in cardiovascular health have occurred concurrently with the evolution of food consumption by humans. Representative examples are the associations of the high-fat and/or high-cholesterol diet [100], high-salt diet [101], Mediterranean diet [102], ketogenic diet [103], and fasting [104], as well as the consumption of red meat [105] or ultra-processed food [106], with cardiovascular health. Nevertheless, how these diet signals are relayed to and converge on cardiovascular health remains poorly understood. Recent advances in nutrient sensing by HSCs provide novel insights into the potential causal link between diet and cardiovascular health. It was found that nutrient metabolism is distinct in HSCs and is tightly coupled with the maintenance and function of HSCs [17].

HSC homeostasis is particularly vulnerable to hypercholesterolemia, which provokes lipid homeostasis perturbation in HSCs featuring lipid droplet accumulation and increased intracellular cholesterol, leading to their biased differentiation into pro-inflammatory myeloid leukocytes. Notably, although cholesterol-lowering treatment can revert the myeloid bias of HSCs, pro-inflammatory traits persist in HSCs [64]. The long-lasting effects of hypercholesterolemia on HSCs imply an involvement of epigenetic memory resembling that induced by sleep problems. Gu et al. report that mechanistically, SREBP2 activation-induced Notch transactivation accounts for HSC expansion in hypercholesterolemia [65]. Our group further reveals that HSCs exhibit a distinct and heterogeneous molecular signature of cholesterol metabolism and demonstrate that a high level of intracellular cholesterol directly promotes the maintenance and myeloid bias of HSCs. Molecularly, the lysosomal transmembrane protein solute carrier family 38 member 9 (SLC38A9), which senses intracellular cholesterol [107], signals to mammalian target of rapamycin (mTOR) to enhance the myeloid differentiation of HSCs and to diminish the ferroptosis vulnerability of HSCs through inhibiting ferritinophagy and upregulating SLC7A11/GPX4 expression. As a result, myeloid-biased HSCs are expanded following the consumption of a high-cholesterol diet (HCD), resulting in the overproduction of myeloid leukocytes [10]. Additionally, the increased HSC proliferation in hypercholesterolemia expedites the somatic evolution and expansion of clones with driver mutations, increasing the risk of clonal hematopoiesis [47]. Furthermore, HCD also promotes BM HSC mobilization and myelopoiesis by perturbing the CXCL12–CXCR4 axis, possibly in association with elevated serum CSF3 levels due to IL-23 generation by splenic dendritic cells and macrophages as well as decreased CXCL12 production by MSCs [66]. Meanwhile, the metabolite of cholesterol 27-hydroxycholesterol can induce HSC mobilization and myelopoiesis via estrogen receptor α (ERα)-mediated CXCR4 downregulation on HSCs [67].

Akin to an HCD, a high-fat diet (HFD) also promotes HSC expansion and differentiation into pro-inflammatory myeloid leukocytes [68,69]. Meanwhile, these alterations persist after the cessation of the HFD, implying an involvement of epigenetic memory [68]. With regard to cellular and molecular mechanisms, Singer et al. suggest that *myeloid differentiation 88* (*MyD88*) activation is implicated in the biased differentiation of HSCs into pro-inflammatory myeloid leukocytes [68]. Lee et al. report that oxidative stress-induced GFI1 upregulation is a pivotal driver of HSC homeostasis alteration following the an HFD [69]. Hermetet et al. report that HSCs express high levels of lipid rafts, while an HFD perturbs the lipid composition of HSCs, thereby disrupting TGF-β receptors within lipid rafts. Downregulated TGF-β/Smad2/3 signaling then promotes the proliferation and myeloid differentiation of HSCs [70]. In addition, the cellular components of the HSC niche are also affected by diet. Most relevantly, an HFD influences adipose tissue, which constitutes 50% to 70% of adult human BM. BM adipocytes play essential roles in regulating HSC homeostasis, which has been reviewed elsewhere [108]. Direct evidence has been established between the consumption of an HFD and the expansion of BM adipocytes [109]. Adipocyte accumulation in BM not only further disrupts the HSC niche through inhibiting bone formation but also impairs HSC homeostasis, both of which are potentially mediated by the overproduction of DPP4 [71]. Additionally, HFD-induced inflammation also exacerbates clonal hematopoiesis [72].

Sodium (Na^+^) intake in the general population is much higher than before and what is currently recommended [110]. It is generally thought that the CVD-promoting effects of a high salt intake are related to water–sodium retention and the resultant hypertension, while recent studies point to an implication of hematopoietic and immunological changes. Excess Na^+^ provokes an inflammatory response through direct effects on innate immune cells and T helper (Th) cells, which have been comprehensively reviewed elsewhere [111]. With this knowledge, Lee et al. suggest that excess Na^+^ may stimulate the maturation of Th17 cells, which would release IL-17 to promote HSC mobilization from BM to the spleen, where they largely produce atherogenic monocytes, consequently accelerating the growth of atherosclerotic plaques [74]. Meanwhile, excess Na^+^ was recently demonstrated to raise intracellular Na^+^ and subsequently interfere with the electron transport chain (ETC) to perturb mitochondrial respiration [75]. Given the critical role of mitochondrial respiration in promoting the activation and myeloid-biased differentiation of HSCs [17], it is tempting to speculate that salt toxicity may involve direct effects on HSC homeostasis.

Inorganic phosphate (Pi) is a common ingredient highly enriched in animal protein but also present in processed foods containing preservatives and additives. Dietary phosphate toxicity is becoming an emerging global health concern [112]. Interestingly, we recently revealed a unique Pi metabolic feature of HSCs manifesting as a high expression level of a Pi transporter, SLC20A1, in HSCs and a high Pi concentration in the BM niche which render HSCs susceptible to hyperphosphatemia induced by a high-phosphate diet (HPD). Elevated intracellular Pi following the consumption of an HPD is sensed by a Pi sensor, PPIP5K2, to augment Akt activation, which further promotes Pi uptake by HSCs through upregulating SLC20A1 as well as drives the expansion and MK/myeloid-biased differentiation of HSCs through boosting GATA2 activity [11]. Additionally, an HPD also increases the apoptosis resistance of HSCs through Akt [76], which may partially underlie the expansion of MK/myeloid-biased HSCs following HPD as well.

The ketogenic diet, which involves consuming high amounts of fats and very low amounts of carbohydrates, has been gaining popularity. The cardiovascular effects of the ketogenic diet remain debatable as it causes rapid and sensible weight loss but raises serum cholesterol [103]. With regard to the hematopoietic compartment, it has been reported that consuming a ketogenic diet increases the circulating levels of free palmitic acid (PA) and PA-associated lipids, which stimulate expansion of HSCs and skew their differentiation into the myeloid lineage, giving rise to pro-inflammatory macrophages in mice [77]. Meanwhile, the pro-inflammation phenotype of macrophages is long-lasting [77], hinting at an implication of epigenetic memory. Regardless, it seems that the ketogenic diet is deleterious to cardiovascular health from the hematopoietic perspective.

On the other hand, diet is a main determinant of gut microbiome diversity [113]. Recently, host microbiota crosstalk has emerged as an essential regulator of the production and functional modulation of hematopoietic and immune cells, with a particular contribution to myeloid leukocyte production. Josefsdottir et al. report that a microbiome is required for HSC maintenance and myelopoiesis, potentially through maintaining a basal level of STAT1 signaling [114]. Meanwhile, NOD1-mediated microbiota sensing in stromal cells regulates HSC maintenance and myelopoiesis through inducing the expression of multiple hematopoietic cytokines [115]. Further, Zhang et al. show that the microbiota can promote the recycling of red blood cells by macrophages in the BM via a short-chain fatty acid butyrate and thereby maintains BM iron availability to dictate the self-renewal and differentiation decisions of HSCs [116]. Moreover, in aged mice, increased levels of microbiota-associated molecular patterns stimulate BM myeloid leukocytes to secrete more IL-1α/β, which directly promotes the myeloid-biased differentiation of HSCs [28]. Inversely, the transplantation of fecal microbiota from young mice can rejuvenate aged HSCs through suppressing inflammation [117]. These lines of evidence suggest that an unhealthy diet pattern may perturb the crosstalk between gut microbiota and HSCs through altering the microbiota composition and metabolism. Indeed, Luo et al. show that HFD induces structural changes in microbiota which are responsible for an alteration in the HSC niche via the activation of peroxisome proliferator-activated receptor γ2 (PPARγ2), leading to impaired osteoblastogenesis but enhanced BM adipogenesis. Consequently, the altered niche contributes to reducing the number and myeloid-biased differentiation of HSCs following the consumption of an HFD [73].

### 4.4. Alcohol Consumption and Tobacco Smoking

There is compelling evidence that alcohol consumption and tobacco smoking cause many diseases, including CVD [118]. Recent studies also include hematopoietic alterations in the deleterious effects of alcohol consumption and tobacco smoking. Acetaldehyde, a metabolite of alcohol produced by alcohol dehydrogenase (ADH), is a toxic and highly reactive substance that reacts violently with intracellular DNA and proteins but can be detoxified by acetaldehyde dehydrogenase (ALDH). Intriguingly, using a mouse model deficient in ALDH2, it was found that within the hematopoietic compartment, only HSCs require ALDH2 for protection against acetaldehyde toxicity. Acetaldehyde accumulation induces DNA double-strand breaks in HSCs, which provoke the apoptosis and proliferation of HSCs, leading to severe HSC attrition in mice [78,79]. Further, the authors showed that acetaldehyde-stressed HSCs share features with aged HSCs including a compromised reconstruction capacity and myeloid-biased differentiation [78,79]. Fortunately, HSCs express *ADH5* to remove endogenous formaldehyde [79], and the DNA-damage-induced *p53* response removes acetaldehyde-damaged HSCs through inducing apoptosis to safeguard HSC pool integrity through restricting the transmission of mutations induced by acetaldehyde [80]. Notably, the authors also characterized the mutational landscape of acetaldehyde-induced DNA damage and show that genomic instability in the form of micronuclei formation occurs in acetaldehyde-stressed HSCs [80]. Micronuclei tend to stimulate the cGAS-STING pathway, and the resultant inflammatory responses may be implicated in the proliferation and myeloid bias of HSCs [40]. However, in rhesus macaque monkeys, chronic voluntary alcohol drinking leads to long-term HSC impairment that is characterized by dampened capacities to proliferate and differentiate into myeloid leukocytes and erythroid cells [119]. The difference between these two animal models may be attributed to the duration and intensity of alcohol/acetaldehyde exposure. Concurrently, chronic excessive alcohol consumption also remodels the HSC niche, thus indirectly modulating HSC homeostasis [81].

It has long been observed that chronic tobacco smoking influences BM hematopoiesis, causing myelocytosis [120]. The main toxic components of tobacco include nicotine, hydrogen cyanide, formaldehyde, and benzene, with nicotine being the primary excitatory substance. Chang et al. report that nicotine exposure increases the frequencies of HSCs and myeloid progenitors in BM, potentially through acting directly on HSCs through nicotinic acetylcholine receptors [82]. Direct evidence linking tobacco smoking to HSC homeostasis alterations remains limited. Researchers are concerned about this problem, and possible mechanisms, such as the remodeling of the HSC niche, have been discussed elsewhere [81,83]. Overall, myeloid-biased hematopoiesis may contribute, at least in part, to the high risk of CVD in individuals who drink and smoke.

### 4.5. Physical Inactivity/Sedentary Behavior

In modern society, living a sedentary lifestyle appears to be becoming a social norm. Physical inactivity/sedentary behavior, defined by not meeting physical activity recommendations, is now recognized as the fourth leading cause of death. The toll of physical inactivity/sedentary behavior is enormous as it is reported to contribute to more than 40 chronic diseases of multiple organs including CVD, metabolic and endocrine diseases, dementia, and cancer [121]. Unfortunately, the deleterious effects of physical inactivity/sedentary behavior remain poorly understood.

Recently, it was shown that the deregulation of HSC homeostasis underlies the limited health span of systemic organs [122,123,124], hinting that HSC homeostasis alterations may link physical inactivity/sedentary behavior to disorders of systemic organs. Indeed, an elegant study demonstrated that sedentary mice have increased numbers of circulating myeloid leukocytes, platelets. and HSCs, with a significantly expanded HSC pool size in BM, compared to exercising mice [84]. Mechanistically, sedentary behavior increases body fat and leptin production in adipose tissue which, by interacting with leptin receptor (LepR)-positive BM stromal cells, decreases the expression of quiescence/retention-promoting niche factors including CXCL12, VCAM1, and ANGPT1. Importantly, sedentary behavior induces HSC epigenome alterations that enhance the chromatin accessibility of genes related to proliferation, myelopoiesis, and B lymphopoiesis. Thus, these hematopoietic changes are long-lasting and persist for 3 weeks in mice following exercise withdrawal. Consequently, the myeloid-biased differentiation of HSCs induced by sedentary behavior aggravates cardiovascular inflammation and outcomes [84]. Recently, Raffin et al. summarized the impacts of sedentary behavior on hallmarks of aging including stem cell exhaustion, cellular senescence, altered intercellular communication, mitochondrial dysfunction, deregulated nutrient sensing, the loss of proteostasis, epigenetic alterations, telomere attrition, and genomic instability [125], all of which are closely associated with HSC homeostasis [12]. Therefore, there may still be a long way to go to comprehensively decipher the impacts of physical inactivity/sedentary behavior on HSC homeostasis and their causal links to CVD.

## 5. Cardiovascular Protective Lifestyle Benefits BM HSC Homeostasis

The deleterious effects of an unhealthy lifestyle are being widely recognized by the public. Fortunately, lifestyle behaviors are modifiable, for instance, through maintaining a healthy mental state, sleeping enough and regularly, ceasing smoking, limiting alcohol consumption, keeping a healthy diet pattern, and engaging in physical exercise. Among these actions, maintaining a healthy diet pattern and physical exercise are the most practicable and cost-effective options in our daily life. Here, we focus on the salutary effects of a healthy diet pattern and physical exercise on BM HSC homeostasis as well as their causal links with cardiovascular health.

### 5.1. Healthy Diet Pattern

A healthy diet is characterized by an appropriate intake of nutrients to meet the physiologic needs of the body without excess consumption [126]. The World Health Organization has recommended a healthy diet pattern that includes balancing energy intake, limiting sugar/salt intake, limiting the consumption of saturated and trans fats while increasing the intake of unsaturated fats, and increasing fruit/vegetable intake. The 2015–2020 Dietary Guidelines for Americans recommend Dietary Approaches to Stop Hypertension, vegetarian diets, and Mediterranean-style and healthy US-style diets as examples of healthy diets [127]. At present, studies refer to the Mediterranean diet in particular as the most cardioprotective due to its abundance of fruits and vegetables [102]. Although the effectiveness of these healthy diets in reducing risk of CVD has been established, the underpinning mechanisms remain largely unknown.

Actually, a healthy diet pattern could benefit BM HSC homeostasis (Table 2). It was recently reported that a higher intake of healthy diet elements (such as vegetables and fruits) than unhealthy elements (such as processed food, red meat, and added salt) reduces the risk of clonal hematopoiesis [128]. Recent studies spotlighted the salutary actions of bioactive compounds such as polyphenols, fiber, and vitamins obtained from an individual’s diet on HSC homeostasis. For instance, vitamin A–retinoic acid signaling is crucial for HSC quiescence as it restricts protein translation as well as ROS and Myc levels. Otherwise, a vitamin A-free diet will cause the proliferation and myeloid-biased differentiation of HSCs [129,130]. Agathocleous et al. report that HSCs have high levels of vitamin C, and systemic vitamin C depletion increases the frequency and induces the myeloid-biased differentiation of HSCs in mice, in part by dampening Tet2 function [131,132]. A form of vitamin B3, nicotinamide riboside, can attenuate age-associated metabolic and functional changes such as the myeloid-biased differentiation of HSCs through restoring mitochondrial metabolic activity and promoting mitochondrial recycling [133,134]. Vitamin E, a widely recognized lipophilic antioxidant, can maintain lipid redox balance in HSCs [135]. In addition, a healthy diet may also favor BM HSC homeostasis through altering gut microbiota composition and/or metabolism [113]. This can be illustrated by urolithin A, which is produced via the gut microbiota-mediated transformation of the natural polyphenols abundant in walnuts and fruits. Girotra et al. report that urolithin A can revert the myeloid bias of aged HSCs through the induction of mitochondrial recycling [136]. Dietary fibers are a complex group comprising lignin and carbohydrates that are enriched in plants and not digested or absorbed in the human body. Dietary fibers were recently proven to offer various health benefits including reducing risk of CVD through modulating the gut microbiota composition and metabolism [137]. Whether BM HSC homeostasis could be safeguarded by dietary fiber through similar mechanisms may be an interesting topic.

Moderate consumption of tea and coffee, the most-consumed beverages worldwide, also shows evidence of cardiovascular protective effects [138]. Previously, we also demonstrated that caffeic acid, a natural hydroxycinnamic acid present in coffee and tea, protects HSCs from oxidative-stress-induced mitochondrial damage [139]. Similarly, tea polyphenols, such as epigallocatechin gallate and theaflavin, are reported to preserve BM HSC homeostasis through attenuating oxidative stress [140,141,142,143]. Whether these effects are implicated in cardiovascular protection via tea and coffee consumption warrants further investigation.

Methods of intermittent and periodic fasting are gaining popularity in weight loss and fitness and also show promise in reducing the risk of CVD [144]. Correspondingly, the salutary effects of fasting on BM HSC homeostasis were observed by Cheng et al. The authors show that prolonged fasting reduces circulating levels of insulin-like growth factor 1 (IGF1) and PKA activity in HSCs, leading to the enhanced stress resistance, self-renewal, and lineage-balanced differentiation of HSCs [145]. Additionally, Jordan et al. report that fasting induces AMPK activation in hepatocytes, which then suppresses the production of monocyte chemoattractant protein-1 (MCP-1/CCL2) via PPARα to reduce the mobilization of pro-inflammatory monocytes [146]. Moreover, short-term fasting also stimulates the HPA axis to release glucocorticoids, which remodel the BM niche as well as the CXCL12–CXCR4 and S1P–S1P1R interactions to promote the retention of monocytes and T cells in BM [147,148]. These effects of fasting on immune cells may also be applicable to HSCs; through them, the mobilization and myeloid bias of HSCs would be alleviated. Notably, the effects of glucocorticoids on HSCs seem to be contradictory between the scenarios of psychosocial stress and fasting. This may result from differences in the durability and intensity of glucocorticoid secretion.

**Table 2 cells-13-00712-t002:** The salutary actions and mechanisms of bioactive compounds derived from a cardiovascular protective diet and fasting in HSC homeostasis.

Bioactive Compound	Influences on BM HSCs	Potential Mechanisms	Species
Vitamin A	Quiescence Balanced differentiation	Restriction of protein translation and levels of ROS and Myc in HSCs through retinoic acid signaling [129,130].	Mouse
Ascorbate	Balanced differentiation	Stimulation of Tet2 activity [131,132].	Mouse Human
Nicotinamide riboside	Balanced differentiation	Restoration of mitochondrial metabolic activity and promotion of mitochondrial recycling [133,134].	Mouse Human
Vitamin E	Maintenance	Promotion of lipid redox balance in HSCs [135].	Mouse
Urolithin A	Balanced differentiation	Induction of mitochondrial recycling [136].	Mouse
Coffee and tea	Maintenance	Promotion of lipid redox balance in HSCs [139,140,141,142,143].	Mouse Human
Fasting	Balanced differentiation Retention	Reduction of circulating levels of IGF-1 and PKA activity in HSCs [145]; AMPK activation in hepatocytes suppresses systemic CCL2 production via PPARα [146]? Remodeling of CXCL12–CXCR4 and S1P–S1P1R interactions in the BM niche [147,148]?	Mouse Human

### 5.2. Physical Exercise

The cardiovascular benefits of physical exercise have been recognized for centuries [149]. Nevertheless, the cellular and molecular mechanisms orchestrating these effects remain poorly understood. In contrast to physical inactivity/sedentary behavior, exercise reduces body fat and the production of leptin by adipose tissue, augmenting quiescence/retention-promoting BM niche factors in LepR^+^ stromal cells to promote quiescence, balanced differentiation, and BM retention of HSCs. Importantly, exercise protects mice and humans with atherosclerosis from chronic myelopoiesis without compromising emergency hematopoiesis, which ensures an efficient defense against infections [84]. Furthermore, exercise also promotes the secretion from BM macrophages of reticulocalbin-2, which activates the cAMP-PKA signaling pathway via its receptor to promote BM fat lipolysis and thereby fuels the lymphoid differentiation of HSCs [150]. Additionally, exercise ameliorates the upregulation of a number of inflammatory pathways associated with old age in HSCs and their niche cells as well as restores aspects of intercellular communication mediated by immune cells [151]. Through the mechanisms described above, the skewed differentiation of HSCs into pro-inflammatory myeloid leukocytes could potentially be reversed by exercise to promote cardiovascular health. Additionally, Shen et al. report that voluntary running increases the frequency of osteolectin^+^ peri-arteriolar stromal cells through the mechanosensitive ion channel PIEZO1. The SCF secreted by osteolectin^+^ peri-arteriolar stromal cells then promotes lymphopoiesis through maintaining early lymphoid progenitors. Notably, in this study, there is no obvious impact of voluntary running on HSC pool size, and HSC differentiation tendency is not investigated [152].

## 6. HSC Plasticity and Its Association with CVD

Notably, HSCs also exhibit the ability to differentiate into non-hematopoietic cells, including cardiovascular cells, which is known as HSC plasticity [153]. From a developmental perspective, embryonic HSCs originate from the hemogenic endothelium through an endothelial-to-hematopoietic transition in the aorta–gonad–mesonephros region. Therefore, HSCs share many cell markers with endothelial cells [154]. Under stress conditions such as radiation injury, BM HSCs also can differentiate into endothelial progenitor cells, which are recruited through circulation to injured blood vessels to promote vascular repair [155]. Additionally, HSCs are also reported to be able to differentiate into adult valve fibroblasts [156]. In recent decades, the role of HSC plasticity in repairing cardiovascular injury has garnered great attention. It has been reported that in patients with refractory peripheral vascular disease or coronary artery disease, mobilizing BM HSCs to circulate to injured sites and the directly injection of HSCs promote local vascular regeneration [157,158]. However, in another HSC infusion experiment on a myocardial infarction model, it was found that circulating HSCs were not recruited at the injury sites yet still caused significant vascular recovery [159]. Furthermore, also in an animal model of myocardial infarction, Jackson et al. report that the transplantation of HSCs can effectively promote the regeneration of injured cardiomyocytes [160]. Mechanistic studies show that HSCs do not differentiate into cardiomyocytes during myocardial infarction [161,162] but promote cardiomyocyte regeneration through cell fusion [163]. Currently, although lifestyle factors have been shown to impact tumor cell plasticity [164], evidence regarding their influences on HSC plasticity is still lacking.

## 7. Conclusions

Despite rapid advances in cardiovascular therapeutics, the CVD burden is on the rise globally. The emerging cardiovascular risk factors arising from the modern lifestyle, including psychosocial stress, sleep problems, physical inactivity/sedentary behavior, an unhealthy diet, alcohol consumption, and tobacco smoking, contribute to this worldwide epidemic. The deleterious effects of these lifestyle factors are chronic and insidious. Although modifiable, lifestyle, as a habit, is often hard to change. Meanwhile, people often overlook the deleterious effects of an unhealthy lifestyle and fail to realize them until disease develops. More unfortunately, how lifestyle signals cardiovascular health remains elusive, with current intervention strategies failing to meet the needs of modern people.

Based on the evidence above, CVD-promoting lifestyles seem to impact HSC homeostasis through direct HSC toxicity or indirectly through neuroendocrine mechanisms, which prompt HSCs to undergo proliferation, expansion, mobilization, MK/myeloid-biased differentiation, and/or pro-inflammatory priming, culminating in the overproduction of pro-inflammatory myeloid leukocytes and platelets, the cellular protagonists of cardiovascular inflammation and thrombosis, respectively. The resultant systemic inflammation may in turn exacerbate the myeloid bias of HSCs, creating a vicious cycle. In contrast, cardiovascular protective lifestyles such as maintaining a healthy diet pattern and physical exercise appear to promote quiescence, BM retention, and the balanced differentiation and/or anti-inflammatory priming of HSCs, which ensures the proper number and function of myeloid leukocytes in circulation and even blunts the age-associated myeloid bias of HSCs, thus conferring cardiovascular protection (Figure 2). Notably, there are interactions between unhealthy lifestyle factors, such as those between psychosocial stress and sleep problems [165] as well as those between sleep problems and unhealthy diet [166], further complicating the interplay between lifestyle, HSC homeostasis, and cardiovascular health. Taken together, we envision that monitoring and preserving hematopoietic health will provide a unique opportunity to safeguard cardiovascular health in the future.

## Figures and Tables

**Figure 1 cells-13-00712-f001:**
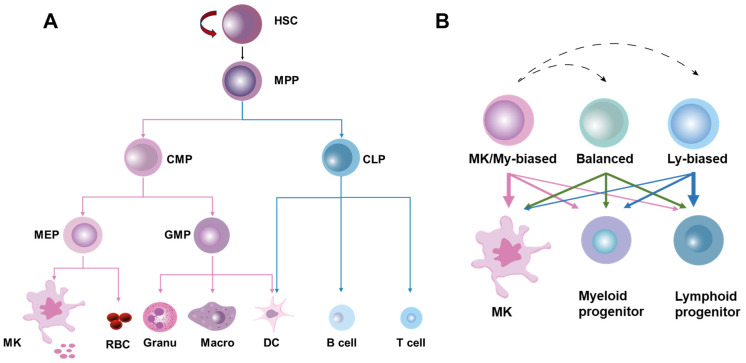
The classical and revised models of hematopoietic hierarchy. (**A**) The classical model of hematopoietic hierarchy, showing that HSCs as a homogeneous population undergo differentiation in a step-by-step manner following the sequence of MPPs, common myeloid/lymphoid progenitors (CMPs/CLPs), lineage-restricted progenitors (GMPs, MEPs, etc.) and mature cells. (**B**) The revised model of hematopoietic hierarchy, showing that HSCs as a heterogeneous population are classified into MK/myeloid-biased, balanced, and lymphoid-biased HSCs on the basis of lineage output.

**Figure 2 cells-13-00712-f002:**
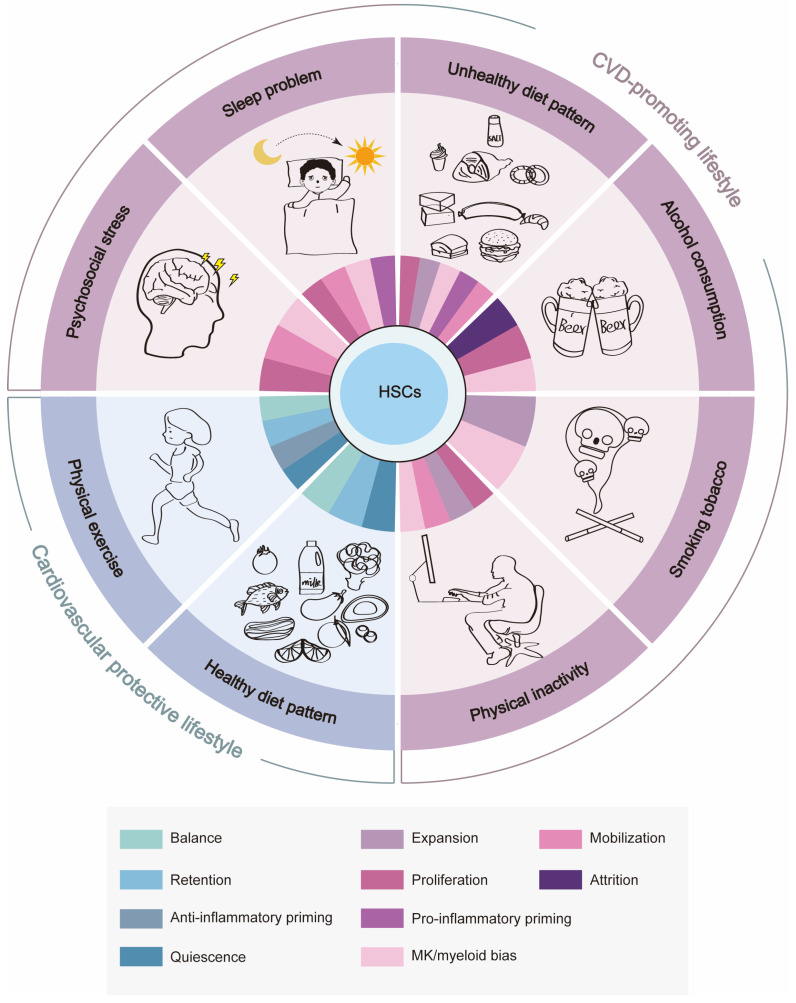
The effects of cardiovascular health-related lifestyle factors on BM HSC homeostasis. CVD-promoting lifestyle factors such as psychosocial stress, sleep problems, an unhealthy diet pattern, alcohol consumption, smoking tobacco, and physical inactivity promote the proliferation, expansion/attrition, mobilization, MK/myeloid-biased differentiation and pro-inflammatory priming of HSCs, while cardiovascular protective lifestyle factors such as a healthy diet pattern and physical exercise promote quiescence, BM retention, balanced differentiation, and the anti-inflammatory priming of HSCs.

**Table 1 cells-13-00712-t001:** The influences and mechanisms of a CVD-promoting lifestyle on HSC homeostasis.

Lifestyle	Influences on BM HSCs	Potential Mechanisms	Species
Psychosocial stress	Proliferation; Mobilization; Myeloid-biased differentiation.	Noradrenaline released by SNS signals niche cells via β-AR signaling to disrupt CXCL12-CXCR4 axis [48,49,50]; dopamine released by SNS activates PKA-Lck-ERK axis in HSCs via D2-type receptor [51]; glucocorticoids released by HPA axis upregulate actin-organizing molecules in HSCs via NR3C1 [21]; decrease in CXCL12 expression in niche cells [52].	Mouse
Sleep problems	Proliferation; Myeloid-biased differentiation; Pro-inflammatory priming; Mobilization?	Less production of hypocretin by hypothalamus promotes CSF1 production by BM pre-neutrophils [53]; brain PGD2 elevation and efflux induce systemic inflammation via DP1 [54]; epigenetic reprogramming and promotion of clonal hematopoiesis through accelerated genetic drift in HSCs [55]; disruption of circadian rhythm leading to deregulated SNS activation [56,57,58], serum proteolytic cascades [59,60,61], and HSC inflammasome signaling [62,63]?	Mouse Human
High-cholesterol diet	Proliferation; Expansion; Mobilization; Myeloid-biased differentiation; Pro-inflammatory priming.	Epigenetic reprogramming in HSCs [64]? SREBP2 activation-mediated Notch upregulation in HSCs [65]; SLC38A9-mTOR axis activation in HSCs [10]; promotion of clonal hematopoiesis through expediting somatic evolution in HSCs [47]; elevated serum levels of CSF3 due to IL-23 generation by splenic dendritic cells and macrophages and decreased CXCL12 production by MSCs [66]; 27-hydroxycholesterol downregulates CXCR4 on HSCs via ERα [67].	Mouse Human
High-fat diet	Proliferation; Expansion; Myeloid-biased differentiation; Pro-inflammatory priming.	*MyD88* activation and epigenetic reprogramming in HSCs [68]? Oxidative stress-induced GFI1 upregulation in HSCs [69]; disruption of TGF-β receptor signaling within lipid rafts of HSCs [70]; expanded BM adipocytes produce excessive DPP4 [71]; inflammation-induced clonal hematopoiesis [72]; structural changes in microbiota alters HSC niche via activation of PPARγ2 [73].	Mouse Human
High sodium intake	Mobilization? Myeloid-biased differentiation?	Increased IL-17 release by Th17 cells [74]? Perturbation of mitochondrial respiration in HSCs [75]?	Mouse
High-Pi diet	Expansion; MK/myeloid-biased differentiation.	Activation of PPIP5K2-Akt axis in HSCs [11]; Akt-mediated increase in apoptosis resistance of HSCs [76].	Mouse
Ketogenic diet	Expansion Myeloid-biased differentiation Pro-inflammatory priming	Increased circulating levels of free PA and PA-associated lipids [77]; epigenetic reprogramming in HSCs [77]?	Mouse
Alcohol consumption	Attrition Proliferation Myeloid-biased differentiation	Acetaldehyde-toxicity-induced DNA damage activates p53 to induce apoptosis [78,79,80]; remodeling of HSC niche [81].	Mouse
Tobacco smoking	Expansion Myeloid-biased differentiation	Nicotine directly acts on nicotinic acetylcholine receptors on HSCs [82]? Remodeling of HSC niche [81,83].	Mouse
Physical inactivity/ sedentary behavior	Proliferation Expansion Mobilization Myeloid-biased differentiation	More leptin is produced by increased body fat and interacts with LepR-positive BM stromal cells to decrease expression of quiescence- and retention-promoting hematopoietic niche factors [84]; epigenetic reprogramming in HSCs [84].	Mouse Human

## Data Availability

Not applicable.

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
