# Peer review of "Hematopoietic Stem Cells as an Integrative Hub Linking Lifestyle to Cardiovascular Health"

_cells, 2024, doi:10.3390/cells13080712_

Round 1

Reviewer 1 Report

Comments and Suggestions for Authors

- please discuss the role of ST2 in this setting. Authors can consider and discuss the papers from Scicchitano P et al. J Clin Med. 2022 May 31;11(11):3142. and Ciccone MM et al. Molecules. 2013 Dec 11;18(12):15314-28. 

- please include a table gathering the main findings from literature

Author Response

Q1: please discuss the role of ST2 in this setting. Authors can consider and discuss the papers from Scicchitano P et al. J Clin Med. 2022 May 31;11(11):3142. and Ciccone MM et al. Molecules. 2013 Dec 11;18(12):15314-28.

Response: Thank you very much for your insightful advice. According to your advice, we have discussed the relevant actions of ST2 in our revised manuscript (highlighted with yellow color in Lines 157-159). The related References (highlighted with yellow color in Lines 728-734) were also added.

Q2: please include a table gathering the main findings from literature

Response: Thank you for your advice. We have provided Tables (Lines 490, 583) in the manuscript.

Reviewer 2 Report

Comments and Suggestions for Authors

In this review authors have tried to generalize the important role of HSC homeostasis and its implication in the Cardiovascular health. This manuscript reads like an opinion rather than a scientific review. The Hematopoietic stem cell plasticity play very important role in Cardiac microenvironement and authors should revisit the review of literature and cite some of the relevant literature to make it more scientifically appealing, informative and directional. 

Comments on the Quality of English Language

This review reads like a newspaper opinion article rather than a scientific review report. I have no specific comment so I requested authors to revisit the review of literature to make it more relevant. 

Author Response

In this review authors have tried to generalize the important role of HSC homeostasis and its implication in the Cardiovascular health. This manuscript reads like an opinion rather than a scientific review. The Hematopoietic stem cell plasticity play very important role in Cardiac microenvironement and authors should revisit the review of literature and cite some of the relevant literature to make it more scientifically appealing, informative and directional.

Response: Thank you very much for your constructive suggestions. To make this review more scientific, we have revisited the cited literatures and revised or removed the overinterpreted statements throughout the manuscript (highlighted with yellow color in Lines 27, 53, 64, 114, 166, 172-173, 182, 228-229, 237-238, 257, 269, 304, 307, 309-310, 318, 325-327, 371-372, 378, 415, 462, 487, 511, 534-535, 574, 576, 625, 631, 633, 641) including Table 1 and Table 2. Meanwhile, we have also cited relevant literatures to make some statements more reliable (highlighted with yellow color in Lines 95, 138, 483).

In addition, according to your advice, we have also added a section reviewing the role of HSC plasticity in CVD (highlighted with yellow color in Lines 585-606). The related References (highlighted with yellow color in Lines 1017-1044) were also added.

Round 2

Reviewer 2 Report

Comments and Suggestions for Authors

Authors have addressed all the previously raised concern and I have no further comments. 

Comments on the Quality of English Language

English looks good